

# Current state of knowledge on *Wolbachia* infection among Coleoptera: a systematic review

Łukasz Kajtoch[1] and Nela Kotásková[2]

[1] Institute of Systematics and Evolution of Animals Polish Academy of Sciences, Krakow, Poland
[2] Faculty of Science, University of Ostrava, Ostrava, Czech Republic

## ABSTRACT

**Background**. Despite great progress in studies on *Wolbachia* infection in insects, the knowledge about its relations with beetle species, populations and individuals, and the effects of bacteria on these hosts, is still unsatisfactory. In this review we summarize the current state of knowledge about *Wolbachia* occurrence and interactions with Coleopteran hosts.

**Methods**. An intensive search of the available literature resulted in the selection of 86 publications that describe the relevant details about *Wolbachia* presence among beetles. These publications were then examined with respect to the distribution and taxonomy of infected hosts and diversity of *Wolbachia* found in beetles. Sequences of *Wolbachia* genes (*16S rDNA, ftsZ*) were used for the phylogenetic analyses.

**Results**. The collected publications revealed that *Wolbachia* has been confirmed in 204 beetle species and that the estimated average prevalence of this bacteria across beetle species is 38.3% and varies greatly across families and genera (0–88% infected members) and is much lower (c. 13%) in geographic studies. The majority of the examined and infected beetles were from Europe and East Asia. The most intensively studied have been two groups of herbivorous beetles: Curculionidae and Chrysomelidae. Coleoptera harbor *Wolbachia* belonging to three supergroups: F found in only three species, and A and B found in similar numbers of beetles (including some doubly infected); however the latter two were most prevalent in different families. A total of 59% of species with precise data were found to be totally infected. Single infections were found in 69% of species and others were doubly- or multiply-infected. *Wolbachia* caused numerous effects on its beetle hosts, including selective sweep with host mtDNA (found in 3% of species), cytoplasmic incompatibility (detected in c. 6% of beetles) and other effects related to reproduction or development (like male-killing, possible parthenogenesis or haplodiploidy induction, and egg development). Phylogenetic reconstructions for *Wolbachia* genes rejected cospeciation between these bacteria and Coleoptera, with minor exceptions found in some Hydraenidae, Curculionidae and Chrysomelidae. In contrast, horizontal transmission of bacteria has been suspected or proven in numerous cases (e.g., among beetles sharing habitats and/or host plants).

**Discussion**. The present knowledge about *Wolbachia* infection across beetle species and populations is very uneven. Even the basic data about infection status in species and frequency of infected species across genera and families is very superficial, as only c. 0.15% of all beetle species have been tested so far. Future studies on *Wolbachia* diversity in Coleoptera should still be based on the Multi-locus Sequence Typing

Corresponding author
Łukasz Kajtoch,
kajtoch@isez.pan.krakow.pl,
lukasz.kajtoch@gmail.com

system, and next-generation sequencing technologies will be important for uncovering *Wolbachia* relations with host evolution and ecology, as well as with other, co-occurring endosymbiotic bacteria.

# INTRODUCTION

The relations between the intracellular $\alpha$-proteobacterium *Wolbachia pipientis* Hertig 1936 (hereafter *Wolbachia*) and its hosts from various groups of arthropods and nematodes have been the object of much research and numerous publications (*O'Neill et al., 1992*; *Werren, Windsor & Guo, 1995*; *Weinert et al., 2015*). The majority of these studies have focused on verifying endosymbiotic bacteria occurrence and diversity in various hosts at different levels: (i) among selected species sharing a geographic area (e.g., *O'Neill et al., 1992*; *Werren, Windsor & Guo, 1995*; 2000), (ii) among species inhabiting the same environment or that are ecologically-associated (e.g., *Stahlhut et al., 2010*), (iii) among species from particular taxonomic groups (e.g., *Czarnetzki & Tebbe, 2004*; *Lachowska, Kajtoch & Knutelski, 2010*; *Sontowski et al., 2015*), and (iv) within populations of selected taxa (e.g., *Stenberg & Lundmark, 2004*; *Mazur et al., 2016*). Another branch of research on the relations between *Wolbachia* and its hosts has focused on host species phylogenetics or population genetics, which is in some cases related to population differentiation and speciation (e.g., *Kubisz et al., 2012*; *Montagna et al., 2014*). In this research, *Wolbachia* is sometimes treated as an additional ''marker''—a source of genetic data about the eco-evolutionary relations of its hosts. A third type of *Wolbachia* studies has concerned the direct or indirect effects of the infection on host fitness, development or survival at the individual and population levels (e.g., *Weeks, Reynolds & Hoffmann, 2002*; *O'Neill, 2007*). Moreover, in a separate branch of research (or in conjunction with the abovementioned types of studies), *Wolbachia* is often examined directly, mainly with respect to strain diversity, distribution and relations with other strains or different co-existing bacteria (*Baldo & Werren, 2007*). All these branches of research have substantially extended the knowledge about the relations between the most widespread intracellular endosymbiont—*Wolbachia* and its various hosts. Moreover, these studies have been expanded to encompass other bacteria with similar biologies and effects on hosts (like *Cardinium, Spiroplasma, Rickettsia*) (*Zchori-Fein & Perlman, 2004*; *Goto, Anbutsu & Fukatsu, 2006*; *Duron et al., 2008*; *Weinert et al., 2015*); however, a great majority of studies are still conducted on *Wolbachia* (*Zug & Hammerstein, 2012*). Recently, the various *Wolbachia* supergroups have been proposed to belong to several ''*Candidatus* Wolbachia'' species (*Ramirez-Puebla et al., 2015*); however, this approach has been criticized (*Lindsey et al., 2016*). Due to the uncertain species status of the ''*Candidatus* Wolbachia'' and because all previous studies considered these presumed different species as distant supergroups, in this review we have followed the previous *Wolbachia* taxonomy.

In summary, *Wolbachia* has been detected in 10–70% of examined hosts (*Hilgenboecker et al., 2008*; *Zug & Hammerstein, 2012*), depending on the geographical, ecological or taxonomical association of the selected species. Moreover, more detailed studies, at the population level, have shown that infection is not as straightforward as was assumed in the early stages of *Wolbachia* research. More and more species have been found to be only partially infected, e.g., in only some parts of their ranges or infection was associated with only some phylogenetic lineages (usually correlated with the distribution of mitochondrial lineages) (*Clark et al., 2001*; *Roehrdanz et al., 2006*). Furthermore, examples of multiply infected species and individuals have been reported, which has important consequences for the understanding of some of the effects of *Wolbachia* infection (*Malloch, Fenton & Butcher, 2000*). *Wolbachia* is known to have numerous effects on its hosts, among which the most interesting and important are those that disturb host reproduction, such as cytoplasmic incompatibility, thelytokous parthenogenesis, feminization of genetic males, male-killing, increased mating success of infected males via sperm competition and the host's complete dependence on bacteria for egg production (for reviews see *Werren, 1997*; *Werren & O'Neill, 1997*; *Stouthamer, Breeuwer & Hurst, 1999*). Some of these effects are responsible for diversification of host populations and consequently *Wolbachia* have probably been involved in speciation (e.g., by the selective sweep of mtDNA or the whole genome of the infected host with the genome of bacteria; *Keller et al., 2004*; *Mazur et al., 2016*). This could be another major factor, additional to those already known, responsible for radiation of insects and particularly beetles.

There are several reviews summarizing the state of knowledge on *Wolbachia* infection among various taxonomic groups of nematodes and arthropods. Over the last years, such reviews have been prepared for the following groups: filarial nematodes (Filarioidea) (*Taylor & Hoerauf, 1999*; *Casiraghi et al., 2001*), crustaceans (Crustacea) (*Cordaux, Bouchon & Greve, 2011*), spiders (Araneae) (*Goodacre et al., 2006*; *Yun et al., 2011*), mites (Acari) (*Chaisiri et al., 2015*), springtails (Collembola) (*Czarnetzki & Tebbe, 2004*), Heteropteran Bugs (Heteroptera) (*Kikuchi & Fukatsu, 2003*), ants (Formicidae) (*Russell, 2012*), wasps (Hymenoptera: Apocrita) (*Shoemaker et al., 2002*) and butterflies (Lepidoptera) (*Tagami & Miura, 2004*). Surprisingly, there is no such review for beetles (Coleoptera), which include large number of diversified taxa, known from various habitats, and whose members belong to all major trophic guilds of animals. Some groups of beetles have been examined with respect to *Wolbachia* infection, but usually only with a limited coverage of species (e.g., weevils, Curculionidae, *Lachowska, Kajtoch & Knutelski, 2010*; leaf beetles; Chrysomelidae, *Clark et al., 2001*; *Jäckel, Mora & Dobler, 2013*; jewel beetles; Buprestidae, *Sontowski et al., 2015* and minute moss beetles, Hydraenidae, *Sontowski et al., 2015*).

In this review we have summarized the current state of knowledge on the relations between beetles and *Wolbachia* by referring to all the abovementioned aspects of research. Moreover, we have highlighted future research directions concerning *Wolbachia* relationships with their diverse Coleopteran hosts.

## SURVEY METHODOLOGY

We searched the scientific literature with Web of Knowledge databases, using the following combination of keywords linked by AND (the Boolean search term to stipulate that the record should contain this AND the next term): "*Wolbachia*" AND "Coleoptera" and "*Wolbachia*" AND "beetles". Our final literature search for this analysis was conducted on December 22, 2017. This produced 322 results. Each result was inspected to determine whether or not it contained information on the subject matter. Articles that had no relevance (e.g., any reports that were not about *Wolbachia*-Coleoptera relations, including those that only had some references to either beetles or bacteria in the citations) were excluded. After the removal of duplicates, 65 were excluded from the remaining articles ($n = 239$) for not being direct reports about *Wolbachia*-Coleoptera relations, 44 were excluded because they examined other hosts and only referred to publications on Coleoptera, and 44 others were excluded because they referred to data already presented in previous publications on Coleoptera. Each document was read critically for the information that it contained on *Wolbachia*-Coleoptera relations, with special reference to answering the study questions listed below. Figure 1 shows a flow diagram for the systematic review following Prisma guidelines (*Moher et al., 2009*). We intended to also use data from The National Center for Biotechnology Information database (GenBank) but the majority of hits (if "*Wolbachia*" AND "Coleoptera" or "beetle" were used) led to either studies not related with *Wolbachia* infection in beetles (which only included references to some other studies on either bacteria or beetles), or to *Wolbachia* sequences submitted to GenBank but without any references to published (and reviewed) articles. Searches in NCBI (GenBank) resulted only in the finding of some beetle hosts, which have been already described in papers found via Web of Science searches.

We examined the collected data on various aspects of *Wolbachia* infection in Coleoptera with respect to the following: the (i) characteristics of the publications (to determine the scope and progress of studies on *Wolbachia*) ($n = 86$), (ii) geographic distribution of infected beetle species and populations ($n = 84$), (iii) sampling design (how many sites and individuals were examined) ($n = 63$), (iv) characteristics of the markers (genes) used for genotyping the bacteria ($n = 82$) and their hosts ($n = 34$), (v) numbers and frequencies of species found to be infected in particular beetle families and genera ($n = 58$), (vi) supergroup prevalence in examined taxonomic groups ($n = 43$), (vii) strain distribution and diversity in populations and individuals ($n = 30$), (vii) effects of *Wolbachia* on its beetle hosts ($n = 39$). Statistical analyses (Spearman correlation for number of publication across years and for the number of examined and number of infected species, Chi$^2$ test for frequency of supergroups and infected taxa in particular taxonomic groups, Chi2 ANOVA for comparison of single/double/multiple infected taxa, Kruskal–Wallis *Z* test for infection frequency in Chrysomelidae and Curculionidae) were done in Statistica 11 (Statsoft).

Finally, we downloaded from GenBank (https://www.ncbi.nlm.nih.gov/genbank/) and the *Wolbachia* MLST database (https://pubmlst.org/wolbachia/) all available sequences of *Wolbachia* genes found in any species of beetle. We restricted further analyses to the most widely used bacteria genes, i.e., *16S rDNA* and cell division protein gene *ftsZ*. Because of
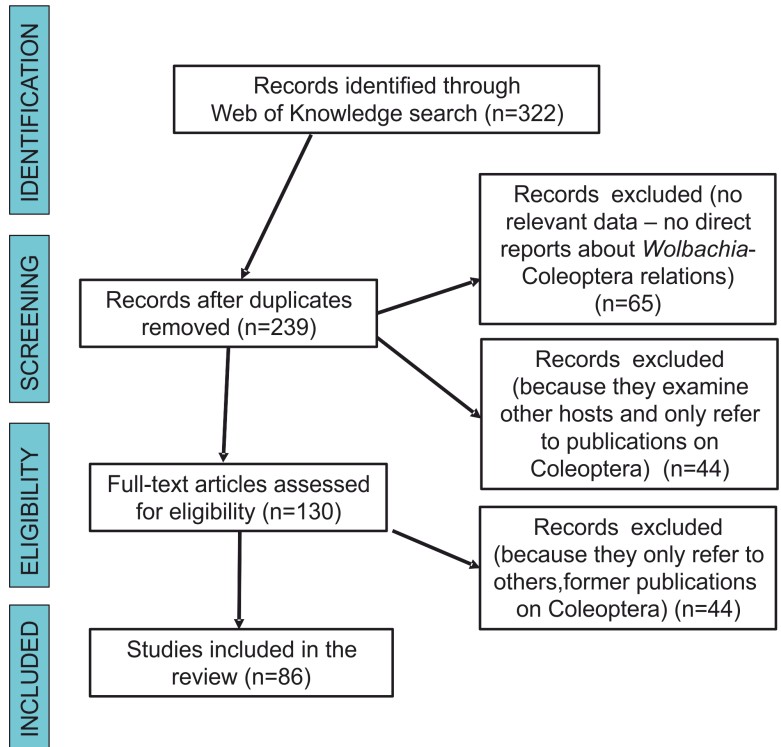

**Figure 1** **Prisma flow-diagram (see *Moher et al., 2009*) for literature on Wolbachia-Coleoptera relations included in this study**

the different lengths and spans of available sequences, the long parts of the 3′ and 5′ ends of each gene were trimmed, which resulted in alignments of length 663 bp for *16S rDNA* and 241 bp for *ftsZ*. The length of the *ftsZ* alignment was particularly short as two different sets of primers have been used for its amplification, and its amplicons only overlapped across a relatively short part of the gene. Phylogenetic trees were only reconstructed for unique gene variants found in particular host taxa. Trees were inferred using Maximum Likelihood (ML) implemented in the IQ-TREE web server (http://www.iqtree.org/) (*Trifinopoulos et al., 2016*) under the following settings Auto selection of substitution model, ultrafast bootstrap approximation (UFBoot) (*Minh, Nguyen & Von Haeseler, 2013*) with 10,000 iterations, maximum correlation coefficient = 0.99, single branch test with use of the approximate Likelihood-Ratio Test (SH-aLRT) (*Anisimova & Gascuel, 2006*; *Guindon et al., 2010*) and other default options.

The nomenclature of host taxa and their systematic positions throughout the paper follow the articles from which the data was derived.

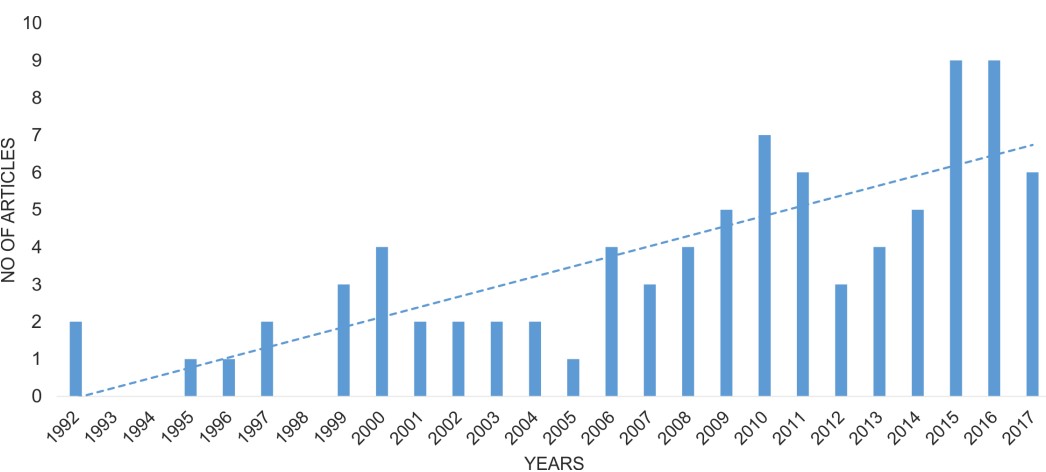

**Figure 2** **Change in the number of publications considering *Wolbachia* infection among Coleoptera.**

# CHARACTERIZATION OF *WOLBACHIA* INFECTION AMONG COLEOPTERA

## Publications

The final list of publications concerning data about *Wolbachia* infection in Coleoptera comprised 86 papers (Table S1). The oldest articles with relevant information about *Wolbachia* infection in beetles were published in 1992 (*Campbell, Bragg & Turner, 1992*; *O'Neill et al., 1992*), and the number of articles since then has increased significantly year by year (Spearman correlation = 0.841; Fig. 2). The majority of these articles (60%) concerned infection in only single beetle species, whereas 19% discussed infection in multiple species belonging to the same genus, 6%—multiple species from the same family, 6%—various species of Coleoptera et al., and a further 9%—studies on geographic groups of insects that included some, usually random species of beetles (*O'Neill et al., 1992*; *Werren, Windsor & Guo, 1995*; *Weinert et al., 2015*.

Most studies were done on Curculionidae (34) and Chrysomelidae (34), following Coccinellidae (10), Tenebrionidae (9), and Sylvanidae (3) (Table S1). The members of all other families were investigated in only 1–2 studies. Consequently, 2.5 and 1.6 Curculionidae and Chrysomelidae species were respectively examined per article. All species of Hydraenidae and Buprestidae were included in only single article (*Sontowski et al., 2015*), whereas limited numbers of species of Coccinellidae and Tenebrionidae were examined in several articles (*Hurst et al., 1999l*; *Hurst et al., 1999b*; *Fialho & Stevens, 1996*; *Fialho & Stevens, 1997*; *Fialho & Stevens, 2000*; *Majerus & Majerus, 2000*; *Weinert et al., 2007*; *Elnagdy et al., 2013*; *Ming et al., 2015*; *Goodacre, Fricke & Martin, 2015*; *Kageyama et al., 2010*; *Li et al., 2015*; *Li et al., 2016b*; *Dudek et al., 2017*). *Wolbachia* infection was only studied more than once in 20 species.

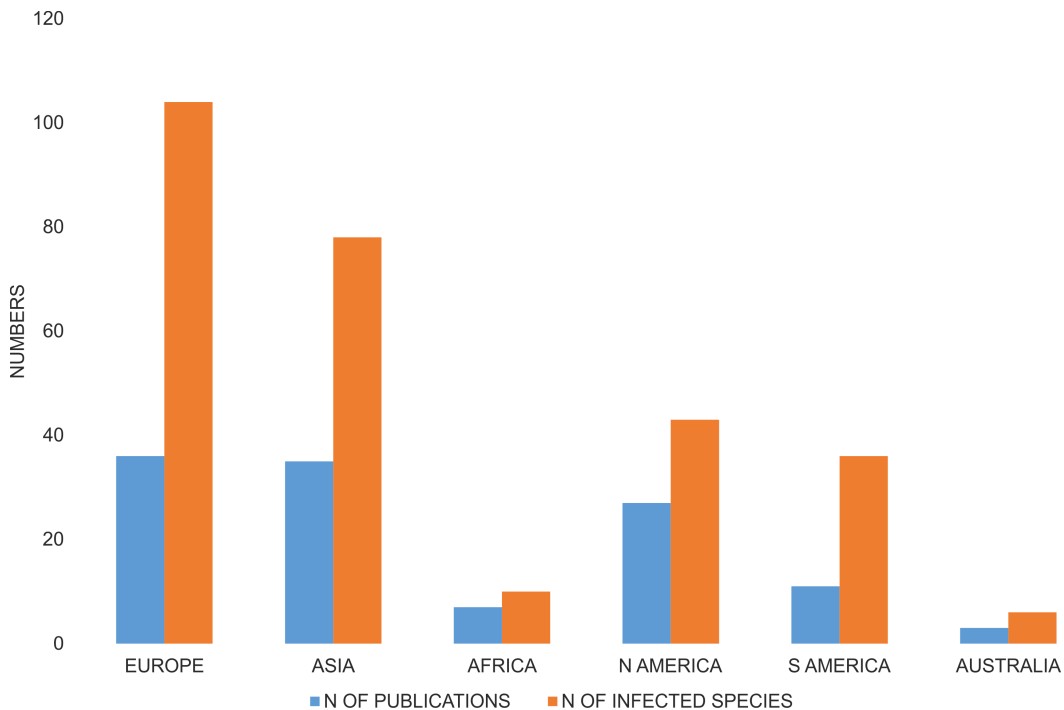

**Figure 3** Number of publications that described *Wolbachia* infection among Coleoptera and number of infected beetle species. Both are shown with respect to the zoogeography of the examined hosts (from which continent the host was collected).

## Sampling design

The majority of species investigated with respect to *Wolbachia* infection were from Europe, and a relatively high number of species were from Asia and both Americas, whereas only ten infected species were from Africa, and three from Australia-Oceania (Fig. 3). A number of publications describing *Wolbachia* infection in Coleoptera had similar geographic coverages (Fig. 3).

Studies were done on samples collected from an average of 5.2 sites and concerned on average 53.0 specimens, or if excluding the most widely studied families Curculionidae and Chrysomelidae, 6.0 sites and 65.1 individuals (Fig. 4). For Curculionidae and Chrysomelidae, these numbers were on average 4.4 and 6.0 sites, respectively, and 40.7 and 70.2 individuals, respectively (Fig. 4). The numbers of sites and individuals examined in particular groups were insignificantly different, with the exception of the numbers of examined individuals in Curculionidae and Chrysomelidae (Fig. 4).

## Examined genetic markers

The most often used *Wolbachia* gene for studies on Coleoptera was *ftsZ*, followed by *hcpA*, *wsp* and *16S rDNA* (Fig. 5). Most studies using *hcpA* also used other MLST genes, including *ftsZ*. On the other hand, many species were only investigated with either *16S rDNA* or *wsp* or *ftsZ* alone. Single studies used *groEL* (*Monochamus alternatus*, Aikawa et al., 2009; *Tribolium madens*, Fialho & Stevens, 2000) or ITS genes (*Tribolium madens*,

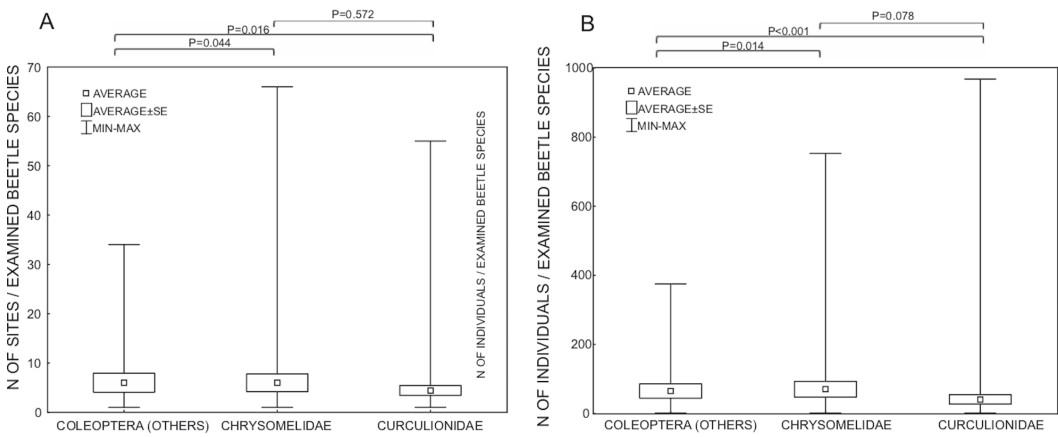

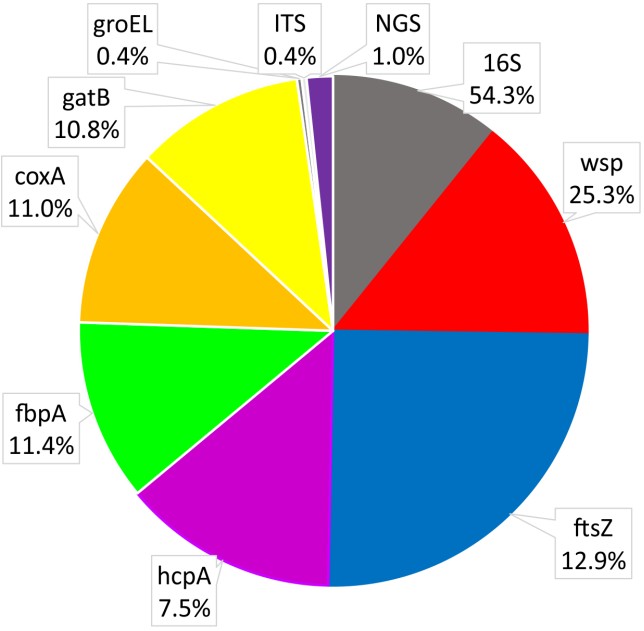

**Figure 4** Number of sites (A) and number of individuals (B) of beetles examined with respect to *Wolbachia* infection. P—Mann-Whitney test *p*-values.

**Figure 5** Shares of *Wolbachia* genes used in studies on *Wolbachia* infection among Coleoptera.

*Fialho & Stevens, 2000*). So far, only five studies have used next-generation sequencing technology (Illumina or 454) to detect *Wolbachia*; two used *16S rDNA* for metabarcoding of microbiota (*Sitona obsoletus, Steriphus variabilis, White et al., 2015; Aleochara bilineata* and *Aleochara* bipustulata, *Bili et al., 2016; Hylobius abietis, Berasategui et al., 2016; Brontispa longissimi, Takano et al., 2017; Harmonia axyridis, Dudek et al., 2017*) and one used shotgun genomic sequencing (*Amara alpine, Heintzman et al., 2014*). For genotyping of hosts, 52.4% of studies utilized fragments of *COI* from mtDNA (usually a barcode fragment of this gene). Fewer studies (23.1%) analyzed *rDNA* (usually *ITS1* and/or *ITS2* spacers), *EF1α* (14.0%),

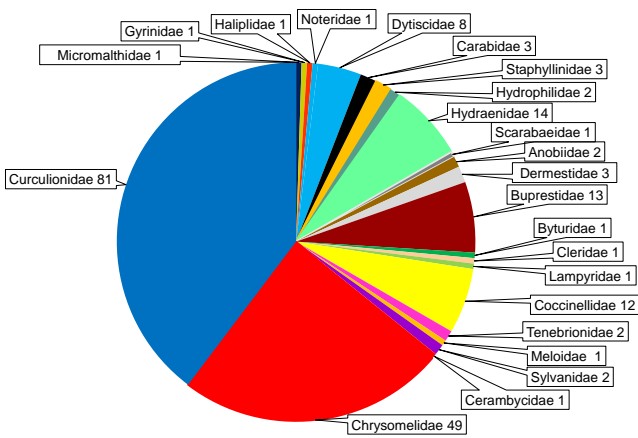

**Figure 6** **Shares of *Wolbachia* infected beetle species across the examined families of Coleoptera.** The numbers presented after the family names indicate the number of infected species.

Wingless (2.2%), Histone H3 (2.2%) and microsatellites (6.1%). In *Wolbachia*-related studies, host genes have been used for several purposes like (i) using host DNA as a control for genetic material quality, (ii) barcoding for host species identification, (iii) phylogenetics, phylogeography and population genetics, (iv) estimating co-evolutionary relations between the bacteria and host, and (v) detecting some of the effects of *Wolbachia* on its hosts (like linkage disequilibrium, selective sweep, cytoplasmic incompatibility).

## Taxonomic coverage

The beetles examined with respect to *Wolbachia* infection belong to 23 families (Micromalthidae, Gyrinidae, Haliplidae, Noteridae, Dytiscidae, Carabidae, Staphyllinidae, Hydrophilidae, Hydraenidae, Anobiidae, Dermestidae, Buprestidae, Byturidae, Cleridae, Lampyridae, Coccinellidae, Tenebrionidae, Scarabeidae, Meloidae, Sylvanidae, Cerambycidae, Chrysomelidae, Curculionidae). In total 204 beetle species were found to harbor *Wolbachia* infection; however, the distribution of infected species among families varied markedly. The highest numbers of infected beetle species were found for the Curculionidae (81 species), Chrysomelidae (49 species), Hydraenidae (14 species), Buprestidae (13 species), Coccinellidae (12 species) and Dytiscidae (8 species) (Fig. 6). In all other families only 1–3 species were reported to harbor *Wolbachia* (Table S1). However, these numbers are biased by the low number of articles (studies) dealing with members of particular beetle families (see above).

Considering infection across beetle genera, the most richly infected genera were *Altica* (Chrysomelidae, 17 species), *Naupactus* (Curculionidae, 11 species), *Hydraena* (Hydraenidae, eight species) and *Agrilus* (Buprestidae, 6 species) (Table S1). In total, 49 genera were found to have infected members (Table S1, Table 1). The infection in Coleoptera was estimated at 38.3% of examined species; however, the proportion of infected species varied greatly between families and genera. At the family level the infection frequency was from 10.5% (Tenebrionidae) to 100% (Noteridae) (*Goodacre, Fricke & Martin, 2015*; *Sontowski et al., 2015*); however when considering only families for which

**Table 1  Share of *Wolbachia* infected species among families and genera of examined beetles.** Only taxonomic groups for which at least two species were tested are presented.

| | N of examined | % of infected | Genus | N of examined | % of infected | genus | N of examined | % of infected |
|---|---|---|---|---|---|---|---|---|
| Family | | | *Barypeithes* | 9 | 11.0 | *Julodis* | 2 | 0.0 |
| Buprestidae | 61 | 23.0 | *Barypeithes* | 9 | 11.0 | *Julodis* | 2 | 0.0 |
| Chrysomelidae | 84 | 45.2 | *Brachysomus* | 4 | 0.0 | *Koreoculio* | 2 | 50.0 |
| Curculionidae | 137 | 41.6 | *Brumoides* | '2 | 0.0 | *Laccophilus* | 2 | 0.0 |
| Dytiscidae | 36 | 16.7 | *Buprestis* | 3 | 0.0 | *Limnebius* | 7 | 28.6 |
| Gyrinidae | 3 | 33.3 | *Byturus* | 3 | 33.0 | *Longitarsus* | 3 | 100.0 |
| Haliplidae | 2 | 50.0 | *Callosbruchus* | 3 | 33.3 | *Meliboeus* | 2 | 0.0 |
| Hydraenidae | 27 | 63.0 | *Callosobruchus* | 7 | 33.0 | *Micraspis* | 2 | 0.0 |
| Hydrophilidae | 12 | 16.7 | *Capnodis* | 3 | 33.3 | *Naupactus* | 16 | 69.0 |
| Noteridae | 2 | 100.0 | *Charidotella* | 2 | 50.0 | *Neoglanis* | 2 | 0.0 |
| Tenebrionidae | 11 | 9.1 | *Chlaenius* | 7 | 14.3 | *Ochthebius* | 12 | 41.7 |
| Subfamily | | | *Chrysobothris* | 3 | 33.3 | *Ophionea* | 3 | 0.0 |
| Bruchinae | 24 | 16.7 | *Coccinella* | 2 | 50.0 | *Oreina* | 5 | 80.0 |
| Galerucinae | 12 | 25.0 | *Crioceris* | 5 | 40.0 | *Otiorhynchus* | 4 | 50.0 |
| Curculionidae | 36 | 16.7 | *Curculio* | 23 | 17.4 | *Paederus* | 3 | 0.0 |
| Scolytinae | 23 | 34.8 | *Cyanapion* | 6 | 50.0 | *Pantomorus* | 3 | 100.0 |
| Genus | | | *Deronectes* | 11 | 45.4 | *Polydrosus* | 4 | 75.0 |
| *Acalymma* | 2 | 100.0 | *Diabrotica* | 12 | 25.0 | *Rhantus* | 2 | 0.0 |
| *Acmaeodera* | 5 | 0.0 | *Dorytomus* | 3 | 67.0 | *Rhinusa* | 3 | 33.3 |
| *Acmaeoderella* | 4 | 0.0 | *Epilachna* | 2 | 0.0 | *Sciaphobus* | 2 | 50.0 |
| *Agabus* | 6 | 16.7 | *Eurymetopus* | 2 | 100.0 | *Sitophilus* | 3 | 100.0 |
| *Agrilus* | 34 | 17.6 | *Gyrinus* | 3 | 33.0 | *Sphenoptera* | 11 | 9.1 |
| *Altica* | 16 | 88.0 | *Haliplus* | 3 | 33.0 | *Strophosoma* | 3 | 67.0 |
| *Anthaxia* | 6 | 16.7 | *Helophorus* | 3 | 0.0 | *Trachypteris* | 2 | 0.0 |
| *Aramigus* | 3 | 100.0 | *Hydraena* | 24 | 33.3 | *Trachys* | 6 | 16.7 |
| *Archarius* | 6 | 16.7 | *Hydroporus* | 5 | 0.0 | *Tribolium* | 8 | 12.5 |
| *Atrichonotus* | 2 | 50.0 | *Hygrotus* | 5 | 20.0 | | | |
| *Aulacophora* | 3 | 0.0 | *Ilybius* | 2 | 0.0 | | | |

more than 30 species were investigated (e.g., *Clark et al., 2001*; *Lachowska, Kajtoch & Knutelski, 2010*; *Rodriguero et al., 2010a*; *Kondo et al., 2011*; *Jäckel, Mora & Dobler, 2013*; *Sontowski et al., 2015*; *Kawasaki et al., 2016*), infection was found in up to 63% of species (Hydraenidae) (Table 1). At lower taxonomic levels, *Wolbachia* was found in 25% of Diabroticite (Chrysomelidae; *Clark et al., 2001*), 14.3–16.7% of Bruchina (Chrysomelidae; *Kondo et al., 2011*), 34.8% of Scolytinae (Curculionidae, *Kawasaki et al., 2016*) and 16.7% of Curculioninii (*Toju et al., 2013*). Among 54 genera in which *Wolbachia* infection was examined for at least two species, 12 genera were completely uninfected, while six genera were completely infected (Table 1). If considering only genera with at least five verified species, *Wolbachia* was found in 0% (*Acmaeodera*; Buprestidae; *Sontowski et al., 2015*) to 88% of species (*Altica*, Chrysomelidae; *Jäckel, Mora & Dobler, 2013*). There was only a marginally negative and insignificant correlation between the number of examined and

number of infected species ($R = -0.078$). If considering only the most widely examined families, Chrysomelidae and Curculionidae, the difference in infection frequency between these two groups was insignificant ($Z = -1.656$, $P = 0.098$). Geographic studies on *Wolbachia* prevalence in insects have found much lower frequencies of infection in Coleoptera species: the bacterium was found in only 10.5% of beetles from Panama and 13.5% of beetles from North America (*Werren, Windsor & Guo, 1995*).

## Wolbachia diversity

Among the various beetle species, *Wolbachia* strains belonged to three supergroups (A, B and F). However, they occurred at very different proportions in different groups of beetles, and these differences were significant ($Chi^2 = 98.78$, $P = 0.000$). Overall, the proportion of beetle species found to be infected with *Wolbachia* strains belonging to supergroups A or B was similar, with approximately 12% of all species harboring either supergroup (either as single infections in different species or populations or as multiple infections within individuals) (Fig. 7), whereas supergroup F was found in only three beetle species: *Agrilus araxenus* and *Lamprodila mirifica* (both Buprestidae; *Sontowski et al., 2015*) and *Rhinocyllus conicus* (Curculionidae; *Campbell, Bragg & Turner, 1992*). In the four groups of beetles with the highest numbers of examined and infected species, the distributions of supergroups varied: in Buprestidae, a similar numbers of species were infected by supergroups A and B (all singly infected), with a relatively high proportion of F infected species (*Sontowski et al., 2015*). In contrast, in Hydraenida, supergroup A dominated over supergroup B (*Sontowski et al., 2015*). This was also the case in Chrysomelidae, with some species infected by both strains (*Kondo et al., 2011*; *Jäckel, Mora & Dobler, 2013*; *Kolasa et al., 2017*). The most varied infections were observed in Curculionidae, with supergroup B dominating, a presence of taxa infected by both A and B supergroups, and a single species infected by F supergroup (*Lachowska, Kajtoch & Knutelski, 2010*; *Rodriguero et al., 2010a*; *Kawasaki et al., 2016*) (Fig. 7). Considering the frequency of infected specimens in the examined beetle species represented by the available data ($n = 106$), 63 species were reported to be totally infected (all individuals possessed *Wolbachia*), whereas 43 species had this bacterium in only some individuals (if exclude Chrysomelidae and Curculionidae: 8 and 15 species, respectively) (Fig. 8). The same calculated for Chrysomelidae resulted in 17 and 10 species, respectively, and for Curculionidae in 38 and 18 species, respectively (Fig. 8). These differences between these values (between these groups of species) were significant ($Ch^2 = 72.03$, $P = 0.000$). A single *Wolbachia* strain was observed in 43 species (species with available data $n = 62$), whereas two strains were reported in 10 species (*Byturus tomentosus*, *Malloch, Fenton & Butcher, 2000*; *Altica quercetorum*, *Jäckel, Mora & Dobler, 2013*; *Callosobruchus chinensis*, *Okayama et al., 2016*; *Chelymorpha alternans*, *Keller et al., 2004*; *Crioceris quaterdecimpunctata* and *Crioceris quinquepunctata*, *Kolasa et al., 2017*; *Adalia bipunctata*, *Majerus & Majerus, 2000*; *Polydrusus inustus*, *Kajtoch, Korotyaev & Lachowska-Cierlik, 2012*; *Cyanapion afer* and *C. spencii*, *Kajtoch, Montagna & Wanat, 2018*) and multiple infection in a further nine species (*Callosobruchus chinensis*, *Kondo et al., 2002*; *Diabrotica barberi*, *Roehrdanz & Levine, 2007*; *Conotrachelus nenuphar*, *Zhang et al., 2010*; *Pityogenes chalcographus*, *Arthofer et al., 2009*; *Xyleborus dispar* and *Xylosandrus*

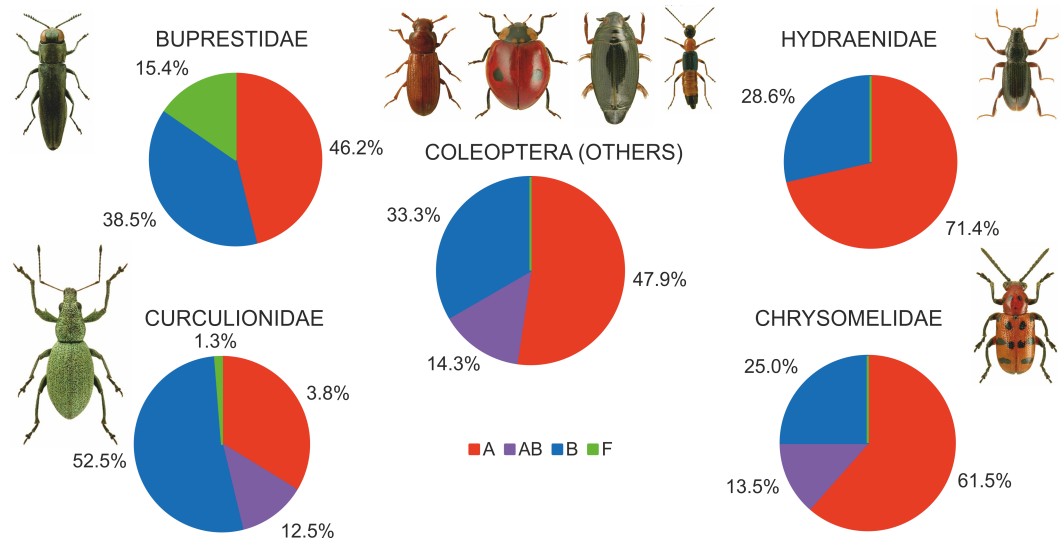

**Figure 7** **Shares of beetles infected by *Wolbachia* supergroups (A, B, F).** (Beetle photographs are from ICONOGRAPHIA COLEOPTERORUM POLONIAE (©Copyright by Prof. Lech Borowiec).

*germanus, Kawasaki et al., 2016*) (Fig. 8). In Chrysomelidae ($n = 22$) these numbers were 12, 5 and five, respectively and in Curculionidae ($n = 37$), 30, 3 and four, respectively (Fig. 8). The numbers of single, double and multiple infected individuals in these groups of beetles differed insignificantly (Chi$^2$ ANOVA $= 2.364$, $P = 0.307$).

## Effects on hosts

*Wolbachia* affected beetle hosts in several ways. Linkage disequilibrium and/or selective sweep between bacteria and host genomes (usually with host mtDNA) were detected in six species (3% or 9% if excluding Chrysomelidae and Curculionidae): two (4%) Chrysomelidae (*Altica lythri, Jäckel, Mora & Dobler, 2013*; *Aphthona nigriscutis, Roehrdanz et al., 2006*) and four (5%) Curculionidae (*Eusomus ovulum, Mazur et al., 2016*; *Naupactus cervinus, Rodriguero, Lanteri & Confalonieri, 2010b, Polydrusus inustus, Polydrusus pilifer, Kajtoch, Korotyaev & Lachowska-Cierlik, 2012*). Cytoplasmic incompatibility was detected or suspected but unconfirmed in 12 (6% or 18% if excluding Chrysomelidae and Curculionidae) Coleoptera: six (13%) Chrysomelidae (*Chelymorpha alternans, Keller et al., 2004, Diabrotica barberi, Roehrdanz & Levine, 2007*, et al., *Diabrotica virgifera virgifera, Giordano, Jackson & Robertson, 1997*; *Callosobruchus chinensis, Kondo et al., 2002*; *Callosobruchus analis, Numajiri, Kondo & Toquenaga, 2017*; *Brontispa longissimi, Takano et al., 2017*), three (4%) of Curculionidae (*Cossomus sp., Zhang et al., 2010*; *Hypothenemus hampei, Mariño, Verle Rodrigues & Bayman, 2017, Xylosandrus germanus, Kawasaki et al., 2016*), one of Sylvanidae (*Oryzaephilus surinamensis, Sharaf et al., 2010*) and one of Tenebrionidae (*Tribolium confusum, Li et al., 2016b; Ming et al., 2015*). Horizontal transfer of *Wolbachia* was detected or suspected in 26 species of Coleoptera (13% or 39% if excluding Chrysomelidae and Curculionidae)—16 (33%) species of Chrysomelidae (several species of *Altica, Jäckel, Mora & Dobler, 2013*,

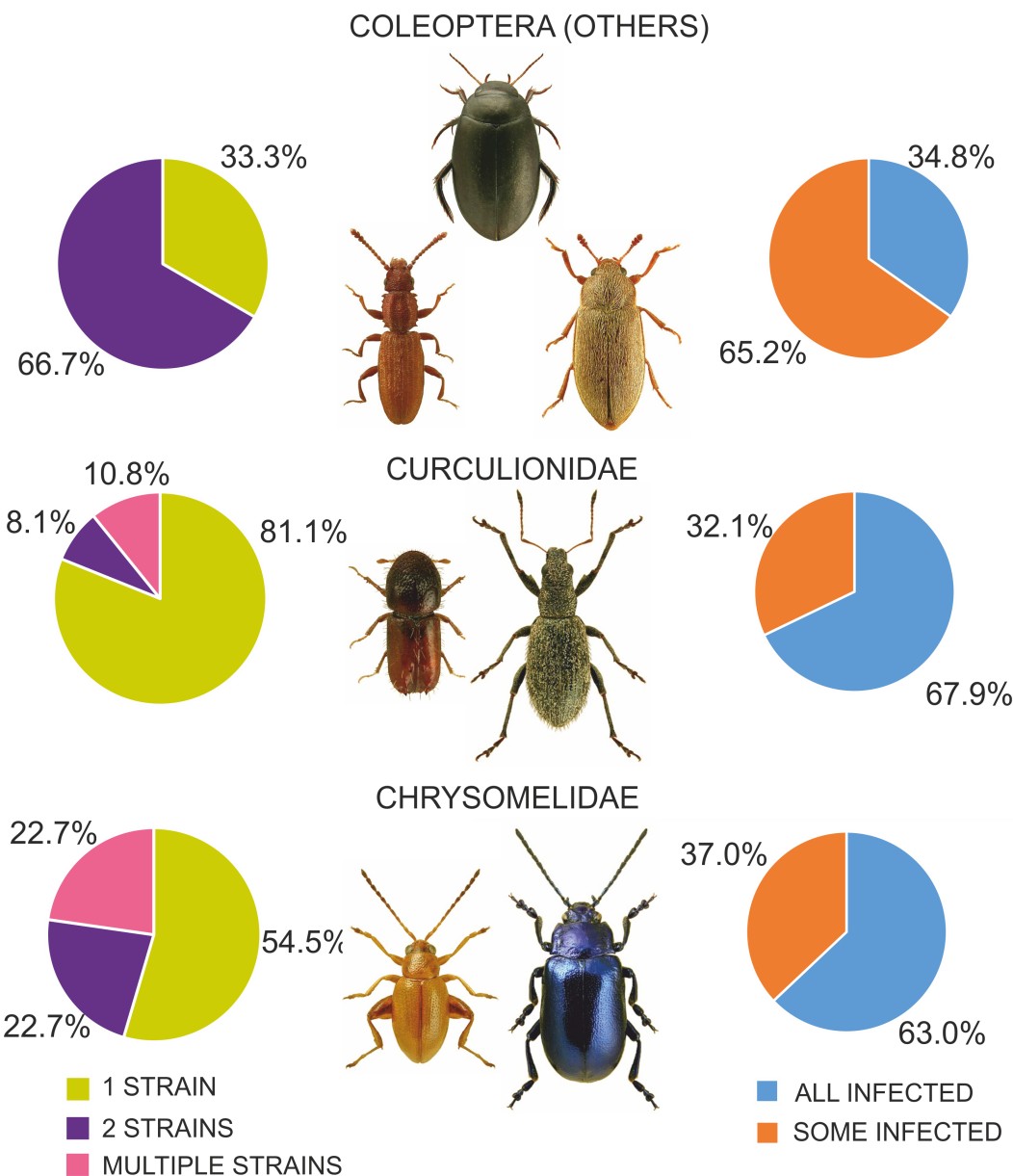

**Figure 8 Diversity of *Wolbachia* infection in Coleoptera with respect to shares of infected individuals within species and numbers of strains found in beetles.** (Beetle photographs are from ICONOGRAPHIA COLEOPTERORUM POLONIAE (©Copyright by Prof. Lech Borowiec).

*Crioceris quaterdecimpunctata* and *Crioceris quinquepunctata*, *Kolasa et al., 2017*) and 10 (14%) species of Curculionidae (members of *Euwallacea*, *Xyleborus*, *Xylosandrus*, *Xyleborinus schaufussi* and *Taphrorychus bicolor*, *Kawasaki et al., 2016*, *Polydrusus* and *Parafoucartia squamulata*, *Kajtoch, Korotyaev & Lachowska-Cierlik, 2012*; *Sitophilus oryzae* and *S. zaemais*, *Carvalho et al., 2014*). Other effects of *Wolbachia* on beetles included the following: (i) transfer of bacteria genes to the autosomes of the host (so far detected only for *Monochamus alternatus*, Cerambycidae, *Aikawa et al., 2009* and

*Callosobruchus chinensis*, Chrysomelidae, *Nikoh et al., 2008*); (ii) coexistence of *Wolbachia* with *Rickettsia* (*Calvia quattuordecimguttata, Coccidula rufa, Coccinella septempunctata, Halyzia sedecimguttata, Rhizobius litura*, *Weinert et al., 2007*; *Sitona obsoletus*, *White et al., 2015*; *Micromalthus debilis*, *Perotti, Young & Braig, 2016*) in the host or with *Spiroplasma* (*Chilocorus bipustulatus*, *Weinert et al., 2007*; *Aleochara bipustulata*, *Bili et al., 2016*) or with both (*Adalia bipunctata*, *Majerus & Majerus, 2000*, *Harmonia axyridis*, *Dudek et al., 2017*; *Curculio sikkimensis*, *Toju & Fukatsu, 2011*; *Aleochara bilineata*, *Bili et al., 2016*); (iii) induction and reinforcement of parthenogenesis, however this effect had weak support and had other possible alternative explanations (numerous species of Naupactini, *Rodriguero et al., 2010a* and *Eusomus ovulum*, *Mazur et al., 2016*; all Curculionidae; *Micromalthus debilis*, *Perotti, Young & Braig, 2016*); (iv) possible induction of haplodiploidy (*Euwallacea interjectus, Euwallacea validus*, Curculionidae *Kawasaki et al., 2016*); (v) male-killing (*Tribolium madens*, Tenebrionidae, *Fialho & Stevens, 2000*); (vi) necessity of infection for egg development (*Otiorhynchus sulcatus*, Curculionidae, *Son et al., 2008*; *Coccotrypes dactyliperda*, *Zchori-Fein, Borad & Harari, 2006*); (vii) populations evolving towards endosymbiont loss and repeated intraspecific horizontal transfer of *Wolbachia* (*Pityogenes chalcographus,* Curculionidae, *Arthofer et al., 2009*), (viii) fitness decline in infected beetles (*Callosobruchus analis*, *Numajiri, Kondo & Toquenaga, 2017*), (ix) modification of sperm (*Chelymorpha alternans*, *Clark et al., 2008*), (x) down-regulation of defense genes in host plants (*Diabrotica virgifera virgifera* on maize, *Barr et al., 2010*).

## Phylogenetic relations

The tree reconstructed for *16S rDNA* included 52 sequences from bacteria found in 45 host beetle species. This tree included three major lineages, with separate clusters of *Wolbachia* sequences belonging to A, B and F supergroups (Fig. S1). F supergroup was represented by a single sequence from *Rhinocyllus conicus* (Curculionidae) (Fig. S1). Sequences assigned to supergroup A (based on information available in the articles) were found to be polyphyletic. Some *16S* sequences from *Xylosandrus* spp. and *Curculio* spp. (Curculionidae), or *Oreina cacaliae* and *Galeruca tanaceti* (Chrysomelidae) clustered as a sister lineage to all other A and B sequences (Fig. S1). Overall, the diversity of *16S* sequences assigned to supergroup B was much greater than those assigned to supergroup A (Fig. S1).

The tree reconstructed for *ftsZ* included 131 sequences found in 114 host beetle species. The *ftsZ* phylogenetic tree resulted in a topology similar to that of *16S rDNA*—it included groups of sequences belonging to A, B and F supergroups (Fig. S2). Supergroup F was represented by *Agrilus araxenus* and *Sphaerobothris aghababiani* (both Buprestidae). Moreover, the supergroup B clade was divided into two clusters, among which one included a small group of sequences found in four beetle hosts: *Chelymorpha alternans* (Chrysomelidae)*, Eurymetopus fallax, Sitophilus oryzae* and *Conotrachelus nenuphar* (all three Curculionidae) (Fig. S2). Also in this gene, the genetic variation of sequences belonging to supergroup A was much lower, and only a few sequences were highly diverged (e.g., strains of *Callosobruchus chinensis*, Chrysomelidae; *Tribolium confusum*, Tenebrionidae or *Polydrosus pilosus,* Curculionidae) (Fig. S2). There was also one slightly

distinct clade that mainly consisted of bacteria sequences found in some Hydraenidae, Curculionidae and Chrysomelidae (Fig. S2).

The abovementioned phylogenetic reconstructions of the relations among *Wolbachia* strains identified on the basis of polymorphism of two genes show that there is no strict correlation between host phylogeny and bacterial strain relationships. Even in studies that covered multiple related species (e.g., those belonging to the same genus), evidence for direct inheritance of *Wolbachia* strains from common ancestors is restricted to Hydraenidae (*Sontowski et al., 2015*) and some species of *Oreina* (*Montagna et al., 2014*) or *Curculio* (*Toju et al., 2013*). In the case of *Altica,* the data show that cospeciation was rare and restricted to a few recently diverged species (*Jäckel, Mora & Dobler, 2013*). In contrast, there are numerous examples of phylogenetically related beetle species possessing different *Wolbachia* strains (e.g., *Lachowska, Kajtoch & Knutelski, 2010*). It is also often the case among related species that some are infected, whereas others not (*Crioceris, Kubisz et al., 2012*; *Oreina, Montagna et al., 2014*; *Cyanapion, Kajtoch, Montagna & Wanat, 2018*); so any assumption that the bacteria were inherited from a common ancestor would also need to consider multiple losses of infection. The latter phenomenon is probable; however, there is no direct evidence from natural populations, at least in studies on beetles, of *Wolbachia* disappearing over time. Some exemplary studies that found *Wolbachia* present in related species, after detailed examination, rejected the idea that bacteria was inherited from a common ancestor. This was because different host species harbored unrelated stains (e.g., among weevils, *Lachowska, Kajtoch & Knutelski, 2010*) or in cases where strains were identical or similar, the hosts were not phylogenetically close to each other (e.g., *Crioceris, Kubisz et al., 2012*). Finally, there is evermore proof of horizontal *Wolbachia* transmission via different mechanisms, such as via predators, parasitoids, common habitat or foraging on the same host plants (*Huigens et al., 2004*; *Stahlhut et al., 2010*; *Caspi-Fluger et al., 2012*; *Ahmed et al., 2015*; *Kolasa et al., 2017*). Studies on beetles have mainly provided indirect evidence of such transmissions. There are known groups of species that inhabit the same environments and share the same or very similar *Wolbachia* strains, e.g., steppic weevils from East-central Europe (*Mazur, Kubisz & Kajtoch, 2014*) and bark beetles in Japan (*Kawasaki et al., 2016*). Recently, evidence for has also appeared for the role of host plants in bacteria spread—*Wolbachia* DNA was detected in two species of *Crioceris* leaf beetles and in their host plant—*Asparagus* spp. (*Kolasa et al., 2017*).

Finally, in light of the proposed ''*Candidatus* Wolbachia'' species, the summarized phylogenetic relations among *Wolbachia* strains infecting various beetles indicate that the taxonomic distinctiveness of supergroups is inconclusive (*Ramirez-Puebla et al., 2015*; *Lindsey et al., 2016*). First, beetles generally harbor members of supergroups A and B, and only occasionally members of supergroup F. Therefore, it is not possible to make any conclusions about broader *Wolbachia* taxonomy based only on *Wolbachia* strains found in Coleoptera. However, there are numerous examples of beetle hosts harboring both supergroups, including beetles in which some *Wolbachia* genes are of supergroup A origin, while others are of supergroup B origin; this indicates that recombination between strains belonging to different supergroups is quite frequent. This is evidence against the

designation of the "*Candidatus* Wolbachia" species, at least with respect to members of supergroup A and B.

## CURRENT GAPS AND FUTURE ENDEAVORS

The present knowledge on *Wolbachia* infection across beetle species and populations is very uneven. Even the basic data about infection statuses in species and frequencies of infected species across genera and families is superficial, as there are only c. 200 beetle species known to be infected. This means that if 38% is the average frequency of infection among beetle species, then only c. 530 species have been tested so far. This is merely c. 0.15% of the total number of beetles, which is estimated to be around 360,000 species (*Farrell, 1998*; *Bouchard et al., 2009*). We know even less at the population level, as the majority of beetle species have only had single individuals tested for *Wolbachia* infection (e.g., *Lachowska, Kajtoch & Knutelski, 2010*; *Sontowski et al., 2015*). These very basic screens have probably underestimated the number of infected species because of false-negative results obtained for species with low or local infection in populations. There is also another and important cause that should be mentioned—low titer infections that are under the detection limit of conventional PCR (e.g., *Arthofer et al., 2009*; *Schneider et al., 2013*). On the other hand, these preliminary estimates could have overestimated the real number infected beetles, as sampling in these studies was rarely random and most often focused on specific groups, e.g., on genera for which preliminary data suggested the presence of *Wolbachia* infection. Indeed, an intensive search of *Wolbachia* infection across hundreds of beetle species from Europe suggested a lower infection rate—c. 27% to be infected (Ł Kajtoch et al., 2018, unpublished data). Also, knowledge about infection at the geographic scale is very uneven, and only Europe and Asia (basically China and Japan) have been relatively well investigated. There is a huge gap in the knowledge for African, Australian and Oceanian beetles, where a high diversity of beetles exists and probably a similar diversity of *Wolbachia* could be expected (e.g., compared to preliminary data available from Central and South America (*Werren, Windsor & Guo, 1995*; *Rodriguero et al., 2010a*).

Little is known about *Wolbachia* diversity in beetle hosts, as the majority of studies used only single genetic markers, and often different genes were sequenced for different taxa. This precludes complex analysis of *Wolbachia* diversity across all tested beetle hosts. This has changed since 2006, since *Baldo et al. (2006)* proposed Multilocus Sequence Typing (MLST), which is based on the genotyping of five housekeeping genes, usually in conjunction with *wsp* sequencing. MLST is and should remain a sufficient way to understand basic *Wolbachia* diversity. On the other hand, to fully understand *Wolbachia* relations among strains and supergroups (or presumed species), between *Wolbachia* and its hosts and especially between *Wolbachia* and other microorganisms, amplicon-sequencing (e.g., *16S* rDNA) or genome-sequencing are needed. This could be achieved thanks to the development of next-generation sequencing technologies (NGS). Surprisingly, despite fast development of NGS in the last years, very few studies have used this technology for studying *Wolbachia* in beetle populations. For example, five studies sequenced *16S* amplicons generated from microbiota and detected *Wolbachia* (*White et al., 2015*; *Bili et*

*al., 2016*; *Berasategui et al., 2016*; *Takano et al., 2017*; *Dudek et al., 2017*). The only study that utilized shotgun sequencing was executed for other purposes and only accidentally showed *Wolbachia* genes in examined species (*Heintzman et al., 2014*). NGS is probably the best prospect for studies on *Wolbachia* infection and diversity, and will help to answer most current riddles and issues.

The big challenge is to understand the impact of infection on beetle biology, physiology and ecology. It is known that *Wolbachia* has several effects on host reproduction, but relatively few studies prove or suggest e.g., cytoplasmic incompatibility, male-killing or other effects on the development of selected beetles (*Clark et al., 2001*; *Keller et al., 2004*; *Roehrdanz et al., 2006*; *Roehrdanz & Levine, 2007*; *Sharaf et al., 2010*; *Zhang et al., 2010*; *Jäckel, Mora & Dobler, 2013*; *Ming et al., 2015*; *Kawasaki et al., 2016*; *Li et al., 2016b*; *Mariño, Verle Rodrigues & Bayman, 2017*; *Numajiri, Kondo & Toquenaga, 2017*; *Takano et al., 2017*). It is very probable that this bacteria has large and frequent effects on beetle reproduction and is consequently partially responsible for beetle radiation, at least in some taxonomic groups, geographic areas or habitats. Also, very few studies have shown data on linkage disequilibrium and selective sweep between bacteriium and host genomes (*Roehrdanz et al., 2006*; *Rodriguero, Lanteri & Confalonieri, 2010b*; *Kajtoch, Korotyaev & Lachowska-Cierlik, 2012*; *Jäckel, Mora & Dobler, 2013*; *Mazur et al., 2016*). These effects could also have probably been involved in speciation of numerous beetles. Moreover, this phenomenon could have serious implications for beetle barcoding, as selective sweep is known to reduce mitochondrial diversity in its hosts and therefore could decrease the number of identified species (*Hurst & Jiggins, 2005*). On the other hand, cytoplasmic incompatibility can lead to the origin of highly diverged phylogenetic mitochondrial lineages within species, which would increase the number of identified taxa (*Smith et al., 2012*). Also here, NGS technologies will enable more sophisticated analyses of these genetic relations and their effects (e.g., by the sequencing of transcriptomes for physiological studies or by genotyping-by-sequencing for phylogenetic studies). Genotyping with NGS should also verify whether the recent assumption that different supergroups are indeed "*Candidatus* Wolbachia" species is correct or not (*Ramirez-Puebla et al., 2015*; *Lindsey et al., 2016*).

Only very preliminary results suggest *Wolbachia* was not only transmitted vertically, but that it could also have spread horizontally (*Jäckel, Mora & Dobler, 2013*; *Carvalho et al., 2014*; *Kawasaki et al., 2016*; *Kolasa et al., 2017*; *Mazur et al., 2016*). Horizontal transmission was considered as an event that happens in evolutionary timescales. Only recently, *Schuler et al. (2013)* showed that such a transfer can happen within a few years after arrival of a new strain. In light of the general lack of cospeciation between bacteria and beetles, horizontal transmission must be a highly underestimated phenomenon. Horizontal transmission of *Wolbachia* among beetles cannot be confirmed without considering other coexisting insects that can mediate transmission, such as predators, parasitoids or beetle prey. Moreover, other arthropods that share habitats with beetles, e.g., phoretic ticks (*Hartelt et al., 2004*) and nematodes (*Casiraghi et al., 2001*), need to be examined. Finally, host plants are promising objects of studies on *Wolbachia* transmission across beetle populations (*Kolasa et al., 2017*), as phloem is probably an important mediator of this bacteria's spread across

insect populations (*DeLay et al., 2012*; *Li et al., 2016a*). Concerning transmission—another very interesting topic is the transfer of *Wolbachia* genes into host genomes (*Dunning Hotopp et al., 2007*; *Koutsovoulos et al., 2014*; *Funkhouser-Jones et al., 2015*). This issue has only been reported twice for beetle hosts so far (*Nikoh et al., 2008*; *Aikawa et al., 2009*). This problem could be important as if such transfers are frequent, simple testing of *Wolbachia* presence in a host based on single or even several gene sequencing could overestimate the number of truly infected species, populations or individuals.

Finally, a very interesting topic for future studies is the examination of the presence of other intracellular and symbiotic bacteria (like *Cardinium, Spiroplasma, Rickettsia*) in Coleoptera and their relations, both with the host and *Wolbachia*. So far, only seven studies have found *Wolbachia* with *Rickettsia* and/or *Spiroplasma* together in beetle hosts (*Majerus & Majerus, 2000*; *Weinert et al., 2007*; *Toju & Fukatsu, 2011*; *White et al., 2015*; *Perotti, Young & Braig, 2016*; *Bili et al., 2016*; *Dudek et al., 2017*). Preliminary results suggest that there is some balance in the number of these bacteria, probably caused by competition within host cells (*Goto, Anbutsu & Fukatsu, 2006*). A recent summary of the presence of these bacteria in insects showed that *Rickettsia* has been found in single species of Micromalthidae, Staphylinidae, Buprestidae, Coccinellidae and Curculionidae (*Werren et al., 1994*; *Lawson et al., 2001*; *Weinert et al., 2007*; *Toju & Fukatsu, 2011*; *White et al., 2015*; *Perotti, Young & Braig, 2016*; *Bili et al., 2016*), *Spiroplasma* in some species of Staphylinidae, Coccinellidae and Curculionidae (*Majerus et al., 1998*; *Hurst et al., 1999a*; *Hurst et al., 1999b*; *Tinsley & Majerus, 2006*; *Weinert et al., 2007*; *Toju & Fukatsu, 2011*; *Bili et al., 2016*), and *Cardinium* has not been detected so far in any beetle species (*Zchori-Fein & Perlman, 2004*). The coexistence of different endosymbiotic bacteria and their effects on hosts should also be investigated with NGS technologies, which are able to detect bacteria in numerous hosts (e.g., individuals) at once and estimate prevalence of bacteria in various hosts or different tissues. NGS has already been proven to be a powerful tool for detecting undescribed bacteria (e.g., it allowed the identification of new Alphaproteobacteria in *Brontispa longissimi*; *Takano et al., 2017*). Different endosymbiotic bacteria could have either similar or contrasting effects on beetle species, populations and individuals and could be the greatest overlooked phenomenon in the evolution and ecology of Coleoptera.

In our opinion, beetles are still an insufficiently examined group of *Wolbachia* hosts, especially considering their systematic and ecological diversity. All issues in studies on *Wolbachia* in Coleoptera are generally the same as in other hosts of these bacteria, or *vice versa*; there is no issue that has been or is being studied on *Wolbachia* infection in other (non-beetle) hosts that could not also be examined in beetle hosts. Also, the extraordinary diversity of beetles (with respect to their diverse systematics at various taxonomic levels, complex phylogenetic relations and extensive ecological relations with each other and numerous other species) makes this group an excellent target for *Wolbachia* studies. The presented summary about *Wolbachia* infection in beetles shows that despite numerous studies, there are still many issues that need to be investigated. We hope that this systematic review will facilitate various future studies on *Wolbachia* infection among beetles.

# ACKNOWLEDGEMENTS

We kindly thank Prof. Lech Borowiec for providing the pictures of beetles from his ICONOGRAPHIA COLEOPTERORUM POLONIAE (©Copyright by Prof. Lech Borowiec, Wrocław 2007–2014, Department of Biodiversity and Evolutionary Taxonomy, University of Wroclaw, Poland), which were used for preparation of the graphics. We are grateful to anonymous reviewers for all their comments and suggestions, which allowed for a great improvement of the manuscript.

## Funding

This work was supported by grant DEC-2013/11/D/NZ8/00583 from the National Science Centre, Poland (to Kajtoch Ł.) and by Institutional Research Support grants (SGS15/PřF/2017) from the University of Ostrava (to Kotásková N.). The funders had no role in study design, data collection and analysis, decision to publish, or preparation of the manuscript.

## Grant Disclosures

The following grant information was disclosed by the authors:
National Science Centre: DEC-2013/11/D/NZ8/00583.
Institutional Research: SGS15/PřF/2017.

## Competing Interests

The authors declare there are no competing interests.

## Author Contributions

- Łukasz Kajtoch conceived and designed the experiments, performed the experiments, analyzed the data, contributed reagents/materials/analysis tools, prepared figures and/or tables, authored or reviewed drafts of the paper, approved the final draft.
- Nela Kotásková performed the experiments, analyzed the data, prepared figures and/or tables, authored or reviewed drafts of the paper, approved the final draft.

## Data Availability

The database of published articles used for systematic review is presented as Table S1 in the manuscript.

## Supplemental Information

Supplemental information for this article can be found online at http://dx.doi.org/10.7717/peerj.4471#supplemental-information.

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

## FURTHER READING

**Chafee ME, Funk DJ, Harrison RG, Bordenstein SR. 2010.** Lateral phage transfer in obligate intracellular bacteria (*Wolbachia*): verification from natural populations. *Molecular Biology and Evolution* **27**:501–505 DOI 10.1093/molbev/msp275.

**Chen SJ, Lu F, Cheng JA, Jiang MX, Way MO. 2012.** Identification and biological role of the endosymbionts *Wolbachia* in rice water weevil (Coleoptera: Curculionidae). *Environmental Entomology* **41**:469–477 DOI 10.1603/EN11195.

**Floate KD, Coghlin PC, Dosdall L. 2011.** A test using *Wolbachia* bacteria to identify eurasian source populations of cabbage seedpod weevil, *Ceutorhynchus obstrictus* (Marsham), in North America. *Environmental Entomology* **40**:818–823 DOI 10.1603/EN10315.

**Frank JH, Erwin T, Hemenway RC. 2009.** Economically Beneficial Ground Beetles. The specialized predators *Pheropsophus aequinoctialis* (L.) and *Stenaptinus jessoensis* (Morawitz): their laboratory behavior and descriptions of immature stages (Coleoptera, Carabidae, Brachininae). *Zookeys* **14**:1–36 DOI 10.3897/zookeys.14.188.

**García-Vázquez D, Ribera I. 2016.** The origin of widespread species in a poor dispersing lineage (diving beetle genus Deronectes). *PeerJ* **4**:e2514 DOI 10.7717/peerj.2514.

**Goryachevaa II, Blekhmanb AV, Andrianova BV, Gorelovaa TV, Zakharova IA. 2015.** Genotypic diversity of *Wolbachia* pipientis in native and invasive *Harmonia axyridis* Pall. 1773 (Coleoptera, Coccinellidae) Populations. *Russian Journal of Genetics* **8**:731–736.

**Heddi A, Grenier AM, Khatchadourian CH, Charles H, Nardon P. 1999.** Four intracellular genomes direct weevil biology: nuclear, mitochondrial, principal endosymbiont, and *Wolbachia*. *Proceedings of the National Academy of Sciences of the United States of America* **96**:6814–6819 DOI 10.1073/pnas.96.12.6814.

**Iwase S, Tani S, Saeki Y, Tuda M, Haran J, Skuhrovec J, Takagi M. 2015.** Dynamics of infection with *Wolbachia* in Hyperapostica (Coleoptera: Curculionidae) during invasion and establishment. *Biological Invasions* **17**:3639–3648 DOI 10.1007/s10530-015-0985-1.

**Jeong G, Kang T, Park H, Choi J, Hwang S, Kim W, Choi Y, Lee K, Park I, Sim H. 2009.** *Wolbachia* infection in the Korean endemic firefly, *Luciola unmunsana* (Coleoptera: Lampyridae). *Journal of Asia-Pacific Entomology* **12**:33–36 DOI 10.1016/j.aspen.2008.11.001.

**Jeyaprakash A, Hoy MA. 2000.** Long PCR improves *Wolbachia* DNA amplification: wsp sequences found in 76% of 63 arthropod species. *Insect Molecular Biology* **9**:393–405 DOI 10.1046/j.1365-2583.2000.00203.x.

**Kawasaki Y, Ito M, Miura K, Kajimura H. 2010.** Superinfection of five *Wolbachia* in the alnus ambrosia beetle, *Xylosandrus germanus* (Blandford) (Coleoptera: Curuculionidae). *Bulletin of Entomological Research* **100**:231–239 DOI 10.1017/S000748530999023X.

**Kittayapong P, Jamnongluk W, Thipaksorn A, Milne JR, Sindhusake C. 2003.** *Wolbachia* infection complexity among insects in the tropical rice-field community. *Molecular Ecology* **12**:1049–1060 DOI 10.1046/j.1365-294X.2003.01793.x.

**Kondo N, Shimada M, Fukatsu T. 1999.** High prevalence of *Wolbachia* in the azuki bean beetle *Callosobruchus chinensis* (Coleoptera, Bruchidae). *Zoological Science* **16**:955–962 DOI 10.2108/zsj.16.955.

**Kondo N, Shimada M, Fukatsu T. 2005.** Infection density of *Wolbachia* endosymbiont affected by co-infection and host genotype. *Biology Letters* **1**:488–491 DOI 10.1098/rsbl.2005.0340.

**Lu F, Kang XY, Lorenz G, Espino L, Jiang MX, Way MO. 2014.** Culture-independent analysis of bacterial communities in the gut of rice water weevil (Coleoptera: Curculionidae). *Annals of the Entomological Society of America* **10**:592–600 DOI 10.1603/AN13145.

**Nirgianaki A, Banks GK, Frohlich DR, Veneti Z, Braig HR, Miller TA, Bedford ID, Markham PG, Savakis C, Bourtzis K. 2003.** *Wolbachia* Infections of the Whitefly *Bemisia tabaci*. *Current Microbiology* **47**:0093–0101 DOI 10.1007/s00284-002-3969-1.

**Pankewitz F, Zöllmer A, Hilker M, Gräser Y. 2007.** Presence of *Wolbachia* in insect eggs containing antimicrobially active anthraquinones. *Microbial Ecology* **54**:713–721 DOI 10.1007/s00248-007-9230-5.

**Piper RW, Compton SG, Rasplus JY, Piry S. 2001.** The species status of *Cathormiocerus britannicus*, an endemic, endangered British weevil. *Biological Conservation* **101**:9–13 DOI 10.1016/S0006-3207(01)00048-9.

**Pourali P, Roayaei Ardakani M, Jolodar A, Razi Jalali MH. 2009.** PCR screening of the *Wolbachia* in some arthropods and nematodes in Khuzestan province. *Iranian Journal of Veterinary Research* **10**:216–222.

**Prakash BM, Puttaraju HP. 2006.** *Wolbachia* endosymbiont in some insect pests of sericulture. *Current Science* **90**:1671–1674.

**Roehrdanz RL, Wichmann SGS. 2013.** *Wolbachia* wsp gene clones detect the distribution of *Wolbachia* variants and wsp hypervariable regions among 160 individuals of a multistrain infected population of *Diabrotica barberi* (Coleoptera: Chrysomelidae). *Annals of the Entomological Society of America* **106**:329–338 DOI 10.1603/AN12118.

**Sharaf K, Hadid Y, Pavliček T, Nevo E. 2013.** Local genetic population divergence in a saw-toothed grain beetle, *Oryzaephilus surinamensis* (L.) (Coleoptera, Cucujidae). *Journal of Stored Products Research* **53**:72–76 DOI 10.1016/j.jspr.2013.03.002.

**Sintupachee S, Milne JR, Poonchaisri S, Baimai V, Kittayapong P. 2006.** Closely related wolbachia strains within the pumpkin arthropod community and the potential for horizontal transmission via the plant. *Microbial Ecology* **51**:294–301 DOI 10.1007/s00248-006-9036-x.

**Toševski I, Caldara R, Jović J, Hernández-Vera G, Baviera C, Gassman A, Emerson BC. 2015.** Host associated genetic divergence and taxonomy in the *Rhinusa pilosa* Gyllenhal species complex: an integrative approach. *Systematic Entomology* **40**:268–287 DOI 10.1111/syen.12109.

**Vega FE, Benavides P, Stuart J, O'Neill SL. 2002.** *Wolbachia* infection in the coffee berry borer (Coleoptera: Scolytidae). *Annals of the Entomological Society of America* **95**:374–378 DOI 10.1603/0013-8746(2002)095[0374:WIITCB]2.0.CO;2.

**Xue H-J, Li W-Z, Nie R-E, Yang X-K. 2011.** Recent speciation in three closely related sympatric specialists: inferences using multi-locus sequence, post-mating isolation and endosymbiont data. *PLOS ONE* **6**:e27834 DOI 10.1371/journal.pone.0027834.