# Peer review of "Current state of knowledge on Wolbachia infection among Coleoptera: a systematic review"

_PeerJ, doi:10.7717/peerj.4471_

## Round 0.1 · original submission · Major Revisions

Dear Drs. Kajtoch and Kotásková:

Thank you for submitting your work to PeerJ. I have now received three independent reviews of your work. The reviewers have all raised concerns to your study, which I believe can be addressed in a thorough revision. Such a revision would likely make your work suitable for publication in PeerJ, thus I encourage you to carefully consider these concerns before undertaking your revision.

In particular, please pay careful attention to the main concern of reviewer 1, which will require you to conduct your literature searches once again. Please note that all your searches should be repeatable as outlined. Thus, you should eliminate any reference to unpublished works, or works that cannot be found on public databases. Seminars, unpublished meeting abstracts, etc. should not be included in your work. Also, note that the reviewers have concerns over your interpretations of previous studies. Try to be objective and not misinterpret the original intentions of the authors. On the other hand, please provide a little more of a summary and general outlook on the literature that you have reviewed. Your own opinions and observations are what will ultimately make your work a valuable resource for the community, which I am sure you are striving to achieve.

There are also many minor suggestions (grammatical mostly) that need to be addressed. As all three reviewers have taken the time to carefully read and comment on this draft, I hope you will also take the time to improve the overall content and writing by addressing these concerns.

Good luck with your revision. I look forward to seeing your revised work.

Best,

-joe

Reviewer 1 ·

Basic reporting

- language ok, but many typos and orthographic errors

Experimental design

- interesting topic
- severe doubts on completeness of data source
- table S1 incomplete

Validity of the findings

- review article, no essentially new findings

Additional comments

Kajtoch and Kotásková present a review about Wolbachia infections in Coleoptera. Similar reviews have, in the last years, been published for other insect species and, while not adding many new facts to the body of literature, they are usually helpful tools for further research. Until now, such a review was missing for beetles.

My main concern here is the integrity of the original dataset. The authors used Google Scholar for literature search. This uncurated, crawler-based search engine is without any doubts extremely valuable for everyday literature recherché, but has some notable limitations: It is notoriously missing important papers, especially if published in the pre-digital era, and, at the same time, adding doubtful or obviously non-scientific sources to the search results. The latter issue may be overcome by careful manual inspection of the search results, the former one not. For these reasons, Web of Science (WoS) has remained the gold standard for scientific literature research. This database has its limitations, too, and is furthermore prohibitively expensive for smaller research institutions, but it is manually curated and covering all SCI journals since more than 100 years.

The authors mention that they retrieved 113 hits by searching for “Wolbachia” AND “Coleoptera”, “Wolbachia” AND, “beetles”, and “Wolbachia” AND “[names of all beetle families, separately]”. It is slightly disturbing that these hits also contain work not found in the database but added by the authors based on not further defined "expert knowledge", and that this material contains non-citable publications like unpublished articles and conference talks. I did a small experiment and searched WoS (retrieved 26 November 2017) for the first two terms the authors have used. "Wolbachia AND coleoptera" gave 164 hits, "Wolbachia AND beetles" 157 hits, and the merging of both lists resulted in 232 papers. While several of these hits would be lost in the manual verification step, I am still sure that the data source used here is highly incomplete.

I have also one important technical remark on the data source as given in Supplementary Table S1: The citations are worthless when no biographic information (year, journal name, volume, page numbers) is provided. This essential information must be added, otherwise the work is unpublishable.

Minor remarks:

- l156: "Maximum likelihood trees were inferred using Maximum likelihood" – well, this does not come as a surprise.
- l159: "substation model" ?
- l343: Is this a "tree", or is it a "network"? Only one can apply.
- l409: Please do not begin sentences with words like "But".
- l411f: NGS is currently revolutionizing molecular ecology and will have its merits in many Wolbachia-related projects yet to come. Anyway, I do not agree that NGS is mandatory to understand Wolbachia diversity. The majority if insect species, and thus Wolbachia strains, is located in countries that, for simple economical reasons, cannot afford NGS in the near to middle future. In contrast, universities in low developed countries are capable of PCR and Sanger techniques, and they sit at the source of large parts of Earth's biodiversity. We should rather encourage such institutions to assess the local biodiversity with simple but efficient techniques like MLST, not discourage them with the NGS overkill.
- l453: Wolbachia gene transfer into the nucleus has been firstly described ten years ago (Dunning Hotopp et al., 2007), and that only two examples are known today is simply not true. Koutsovoulos, Makepeace, Tanya, & Blaxter (2014) and Funkhouser-Jones et al. (2015) are two quite recent examples out of several others. It is shameful when a literature review article misses a broad body of published research.
- Finally, the paper contains a lot of typos, false uses of the comma, missing or additional spaces, and some orthographic errors. It should be copy-edited carefully.

References

Dunning Hotopp, J. C., Clark, M. E., Oliveira, D. C. S. G., Foster, J. M., Fischer, P., Muñoz Torres, M. C., … Werren, J. H. (2007). Widespread lateral gene transfer from intracellular bacteria to multicellular eukaryotes. Science (New York, N.Y.), 317(5845), 1753–1756. doi:10.1126/science.1142490
Funkhouser-Jones, L. J., Sehnert, S. R., Martínez-Rodríguez, P., Toribio-Fernández, R., Pita, M., Bella, J. L., & Bordenstein, S. R. (2015). Wolbachia co-infection in a hybrid zone: discovery of horizontal gene transfers from two Wolbachia supergroups into an animal genome. PeerJ, 3, e1479. doi:10.7717/peerj.1479
Koutsovoulos, G., Makepeace, B., Tanya, V. N., & Blaxter, M. (2014). Palaeosymbiosis Revealed by Genomic Fossils of Wolbachia in a Strongyloidean Nematode. PLOS Genetics, 10(6), e1004397. doi:10.1371/journal.pgen.1004397

Reviewer 2 ·

Basic reporting

see below

Experimental design

see below

Validity of the findings

see below

Additional comments

Review for Peerj-21704

The paper by Kajtoch & Kotásková is a nice summary of current knowledge on Wolbachia in beetles. Wolbachia is arguably the most common, and may be the most important endosymbiont of arthropods. There are a lot of studies on Wolbachia in many host taxa, and especially for people new to this field, the large number of studies may be overwhelming. Reviews on endosymbionts of particular host taxa such as the one presented here are therefore very helpful and represent valuable resources for the community.

I want to commend the authors for the effort put into compiling this dataset, which is mostly clearly presented and illustrated. The only major issue I see with this manuscript is the lack of any synthesis that goes beyond summarizing the findings of previous studies. It would be interesting to know if, in the authors opinion, beetles are peculiar Wolbachia hosts in any respect, or if there is something especially noteworthy that only seems to apply to beetles and not any other Wolbachia hosts. Furthermore, everything that is mentioned in the “outlook” section of the manuscript applies to Wolbachia studies in general and is not specific to Wolbachia-beetle interactions.

There are other -mostly minor- issues that should be addressed in a revision. Please find details on these below, hopefully constructive.

Line 81ff:
As potential addition to this list, there are reviews on mites (Chaisiri et al. 2015) and ants (Russell 2012) as well. The latter one also compares Wolbachia incidence between many different arthropod groups, and might be worth having a look for inspiration.

Line 88:
“[beetles] are the most species rich organisms on earth […]” – One often finds similar statements, but they do not make sense at all. Insects are even more species rich than beetles, and arthropods are more species rich than insects. This is just to illustrate that all phylogenetic ranks (except maybe species) are artificial, and therefore those statements should be avoided.

L109–112:
I don’t know of any study showing that Wolbachia is “responsible” for speciation, if the authors claim this here, it should be referenced with an example.

L120–121:
As far as I know, google searches always use the boolean “AND”, regardless if it is typed in the search field.

L126:
I searched my personal literature database and found a number of papers about Wolbachia in beetles that are not mentioned in your manuscript (Pankewitz et al. 2007; Chafee et al. 2010; Kageyama et al. 2010; Toju & Fukatsu 2010; Xue et al. 2011; Toju et al. 2013; Iwase et al. 2015). I am unsure whether this is due to these things not showing up in your search because the papers used are not listed anywhere separately. I would suggest to provide a list of all the papers used and, should you have missed the papers mentioned above, re-design your search strategy to accomodate potential shortcomings that have led to the oversight of these.
Another way to discover potentially overlooked Wolbachia-beetle associations would be to check the NCBI nucleotide database for Wolbachia sequences from beetles. Maybe also worthwhile is to search the databases used by previous meta-analyses investigating global Wolbachia incidences (Weinert et al. 2007; Hilgenboecker et al. 2008; Russell 2012).

L178–179
Please clarify or rephrase.

L198
From here on, the figure numbering seems to be wrong. Also, Figure 2 (“ Change in the number of publications considering Wolbachia infection among Coleoptera”) is not mentioned anywhere.

L229ff
The numbers here and elsewhere in the manuscript would be more meaningful if they where corrected for the number of known species and the number of species tested. E.g., with 8 Wolbachia- positive species found, Hydraena is listed as one of the most “richly infected genera”. However, this is a huge genus, with at least a few hundred described species, so what does a finding of 8 infected species mean? I think in a review like this, these numbers should be put in perspective.

L294ff
Were these cases of CI experimentally determined?

L319ff
I am unsure how helpful the phylogenetic analyses are. What is the aim here? The discussion doesn’t really go beyond supergroup designation, and this is already discussed in the original publications. Maybe think about omiting this part or moving it into the supplement.

L327
“Intermediate between supergroup F and other supergroups” is not really a valid description of a phylogenetic tree.

L380
Suggest to change “proof” to “evidence for”

L397–398
Which search is meant here? Is there a reference missing?

L439–440
Please elaborate on how this would help in species designations in Wolbachia, this is not obvious to me.

Fig. 3
Maybe think about replacing the number of infected species with a stacked bar showing the number of tested/infected species.

Fig. 4
It looks like you are comparing all Coleoptera vs two beetle families. If the families are included in the Coleoptera counts, this would not be valid. If that is not the case, please call this group it “other Coleoptera” or similar for clarification. Also, please provide more details in the Figure legend.

Fig. 5, 6, 8, 9
This is more a matter of personal taste, but it was repeatedly argued that pie charts are very difficult to interpret (especially 3D pie charts).

Fig. 7
As mentioned above, relative proportions would be more insightful here.

Fig. 9
This is a bit hard to follow. What is “some infected” and “all infected”?


References

Chafee ME, Funk DJ, Harrison RG, Bordenstein SR (2010) Lateral Phage Transfer in Obligate Intracellular Bacteria (Wolbachia): Verification from Natural Populations. Molecular Biology and Evolution 27, 501–505.

Chaisiri K, McGarry JW, Morand S, Makepeace B (2015) Symbiosis in an overlooked microcosm: a systematic review of the bacterial flora of mites. Parasitology FirstView, 1–11.

Hilgenboecker K, Hammerstein P, Schlattmann P, Telschow A, Werren JH (2008) How many species are infected with Wolbachia?- a statistical analysis of current data. Fems Microbiology Letters 281, 215–220.

Iwase S, Tani S, Saeki Y, Tuda M, Haran J, Skuhrovec J, Takagi M (2015) Dynamics of infection with Wolbachia in Hypera postica (Coleoptera: Curculionidae) during invasion and establishment. Biological Invasions , 1–10.

Kageyama D, Narita S, Imamura T, Miyanoshita A (2010) Detection and identification of Wolbachia endosymbionts from laboratory stocks of stored-product insect pests and their parasitoids. Journal of Stored Products Research 46, 13–19.

Pankewitz F, Zollmer A, Hilker M, Graser Y (2007) Presence of Wolbachia in insect eggs containing antimicrobially active anthraquinones. Microbial Ecology 54, 713–721.

Russell JA (2012) The ants (Hymenoptera: Formicidae) are unique and enigmatic hosts of prevalent Wolbachia (Alphaproteobacteria) symbionts. Myrmecological News 16, 7–23.

Toju H, Fukatsu T (2010) Diversity and infection prevalence of endosymbionts in natural populations of the chestnut weevil: relevance of local climate and host plants. Molecular Ecology 20, 853–868.

Toju H, Tanabe AS, Notsu Y, Sota T, Fukatsu T (2013) Diversification of endosymbiosis: replacements, co-speciation and promiscuity of bacteriocyte symbionts in weevils. Isme Journal 7, 1378–1390.

Weinert LA, Tinsley MC, Temperley M, Jiggins FM (2007) Are we underestimating the diversity and incidence of insect bacterial symbionts? A case study in ladybird beetles. Biology Letters 3, 678–681.

Xue HJ, Li WZ, Nie RE, Yang XK (2011) Recent Speciation in Three Closely Related Sympatric Specialists: Inferences Using Multi-Locus Sequence, Post-Mating Isolation and Endosymbiont Data. Plos One 6, .

Reviewer 3 ·

Basic reporting

The review on Wolbachia in Coleoptera summarizes the data obtained from a thorough literature search and nicely illustrates the basic results by pie charts. As such the paper is a very nice resource for future studies. I am somewhat concerned, however, that the authors are not always sufficiently careful not to overstretch the conclusions of the original papers. To give two examples: I reread two of the heavily quoted papers, Mazur et al. 2016 and Jäckel et al. 2013, and don't agree that Mazur et al. unambigiously demonstrated that Wolbachia induced or reinforced parthenogenesis in Eusomus ovulum (l. 311 - 313) - it could also be that Wolbachia infected the beetles when they were already parthenogenetic. The Jäckel paper on the other hand talks about the possible connection between CI and selective sweeps, but does not unambigiously demonstrate that this was the case in the Altica beetles as the authors claim (l. 294-295). Especially since people tend to quote from reviews, the authors need to be extremely careful to correctly quote the results.
While the structure of the paper is mostly straight forward I feel that the introduction should be restructured. The general information on Wolbachia effects given in l. 95-l. 113 should be moved further up and substantiated with the best quotes across all insects, not just with those from the beetle literature. The aim of the study, i.e. compiling the data for beetles l. 81-94, should be the last part of the introduction.
When the authors come back to the idea of splitting Wolbachia into several species (l. 440-442), they should discuss at least roughly in how far their phlylogenetic analyses agree or do not with the proposed distinctions
The English would benefit from editing by a native speaker, e.g.
l. 233 and all pie charts - replace 'share' by 'proportion'
l. 236 and 243 replace 'verified' by 'investigated'
l. 244 replace 'members' by 'of the species'
l. 256 rearrange to 'supergroups A and B' and l. 259 to 'supergroup F'
l. 270 replace 'within' by 'represented by'
and many more instances where an English correction would help.

Experimental design

o.k.

Validity of the findings

see above

---

## Round 0.2 · Minor Revisions

Dear Drs. Kajtoch and Kotásková:

Thanks for submitting your revision to PeerJ. I believe your work is much improved, however, there are still some concerns that came up regarding the revision. Please spend some time with the three reviews, and please pay particular attention to the comments of reviewers 1 and 3. I would like to see the outstanding host associations included in your final study, as reviewer 1 has gone to great lengths to help you. I also agree with reviewer 3 regarding the use of wsp for phylogeny estimation. I realize it may be a lot of work to re-estimate a phylogeny on a different molecule(s), but perhaps you can do this, or at least remove (or move the tree to the supplement) the tree from the manuscript.

There are many helpful suggestions here, and I firmly believe if you entertain most of these your work will be suitable for publication in PeerJ.

Thanks for considering another revision, and I do hope I see your new manuscript very soon.

Best,
-joe

Reviewer 1 ·

Basic reporting

English very much improved, references mostly fine, well structured.

Experimental design

Review article, no new findings. Serach strategy for literature now state of the art, inclusion criteria now well defined.

Validity of the findings

Data robust, conclusions clear, except for the wsp phylogeny - see below.

Additional comments

The authors have implemented most of the issues raised by the reviewers and massively improved the ms. Before I can recommend an accept, a few points should still be addressed.

Major issues:

l 78f: Several attempts have failed to reproduce the findings of Jeyaprakash and Hoy (2000), and, although never retracted, most Wolbachia researches tend to not citing this reference anymore. The current valid reference for Wolbachia abundance is Zug and Hammerstein (2012).
l 142: Please provide details which statistical tests have been used and where.
l 355f: It is not valid to use wsp for phylogenetic reconstruction, see Baldo et al. (2005). I would like most if this phylogeny is simply removed – the ms would lose in fact nothing from it. If the authors decide not to remove it, they have, anyway, to discuss why it is invalid. And by the way, it is still not a network!
l 416: I absolutely agree on the causes of Wolbachia underestimation, but there is another and important cause that should be mentioned: Low titer infections that are under the detection limit of conventional PCR, see, e.g., Arthofer et al. (2009), Schneider et al. (2013).
l 458: "led to speciation" is very strong, until now there is not a single, hard evidence that Wolbachia alone can split a species into two. Tone down to something like "have probably been involved in speciation".
l 469f: Horizontal transmission was considered as an event that happens in evolutionary timescales. Only recently, Schuler et al. (2013) showed that such a transfer can happen within a few years after arrival of a new strain – I think this should be mentioned here.

Minor issues:

l 41: Testing and examining are in fact the same. Please remove "and/or examined".
l 42: Add comma before "and".
l 61: Add comma after "research".
l 87: One dot too much in the reference.
l 221: Insert comma after "however".
l 244: Insert "of" before "species".
l 247: Replace colon with comma.
l 278: Missing space after "tomentosus,".
l 285: Insert comma after "22)".
l 307: Missing space after "Euwallacea,".
l 379: was -> were
l 432: Remove comma after "et al.".
l 443: Tautology, remove "single".
l 444: also -> only
l 469: suggested -> suggest
l 473: Remove comma.
l 487: Remove comma.
l 488f: The authors talk about "only three studies" but have seven papers in the according reference; can it be that this is misplaced?

References

Arthofer W, Riegler M, Avtzis DN, Stauffer C. 2009. Evidence for low-titre infections in insect symbiosis: Wolbachia in the bark beetle Pityogenes chalcographus (Coleoptera, Scolytinae). Environ. Microbiol. 11:1923–1933.
Baldo L, Lo N, Werren JH. 2005. Mosaic Nature of the Wolbachia Surface Protein. J. Bacteriol. 187:5406–5418.
Jeyaprakash A, Hoy MA. 2000. Long PCR improves Wolbachia DNA amplification: wsp sequences found in 76% of sixty-three arthropod species. Insect Mol. Biol. 9:393–405.
Schneider DI, Riegler M, Arthofer W, Merçot H, Stauffer C, Miller WJ. 2013. Uncovering Wolbachia Diversity upon Artificial Host Transfer. PLoS ONE 8:e82402.
Schuler H, Bertheau C, Egan SP, Feder JL, Riegler M, Schlick-Steiner BC, Steiner FM, Johannesen J, Kern P, Tuba K, et al. 2013. Evidence for a recent horizontal transmission and spatial spread of Wolbachia from endemic Rhagoletis cerasi (Diptera: Tephritidae) to invasive Rhagoletis cingulata in Europe. Mol. Ecol. 22:4101–4111.
Zug R, Hammerstein P. 2012. Still a Host of Hosts for Wolbachia: Analysis of Recent Data Suggests That 40% of Terrestrial Arthropod Species Are Infected. PLoS ONE 7:e38544.

Reviewer 2 ·

Basic reporting

see below

Experimental design

see below

Validity of the findings

see below

Additional comments

Admittedly, I am a bit disappointed by this revised version of the ‘Wolbachia in beetles review’ of Kajtoch & Kotásková, as they have chosen to ignore several of my suggestions. I do understand that responding to reviewer’s comments can be tedious and sometimes annoying, but just as I haven taken the time to carefully review this paper, I wished the authors had taken the time to carefully address my comments. This does not at all mean they have to agree with everything I said, but I think its fair to expect a response.

Specifically, I suggested to check the NCBI Genbank database for potentially overlooked beetle-Wolbachia associations. I further suggested to check previously databases of Wolbachia-host associations that were compiled by authors of other meta-analyses, such as the one by Weinert et al (2015, http://doi.org/10.1098/rspb.2015.0249). I did the suggested searches and found an additional 184 (!) genera of beetles that were reported to harbour Wolbachia in the literature, but are missing from the current paper (the list is attached below). Especially the search in the Weinert et al database would have been very straightforward, as the authors have provided their dataset as supplementary file to their paper (freely accessible here: http://rspb.royalsocietypublishing.org/highwire/filestream/66583/field_highwire_adjunct_files/0/Database.xlsx). As the current paper aims to summarize all current knowledge on Wolbachia in beetles, I suggest that the authors check these missing entries and the accompanying papers, and include the valid reports to their list of Wolbachia infected species. Each of the genera should be in either the Weinert et al database or retrievable from NCBI’s Genbank through ‘genus AND Wolbachia’ searches.

Other, minor comments:

L41–42 ‘Future studies on Wolbachia diversity in Coleoptera should still be based on the Multi-locus Sequence Typing system’ – why? This is not really discussed in the paper.

L90–91 ‘Some of these effects are responsible for […] speciation […]’ – There is no evidence for that (or if there is, please cite). I already mentioned this in first my review.

L106–107 ‘[…] beetles (Coleoptera), which are the most species rich and diversified group of organisms on Earth […]’ – This doesn’t make sense. I already mentioned this in first my review.

L144 I am puzzled by the fact that my search on NCBI’s Genbank revealed many additional Wolbachia- beetle associations (see above).

L228ff I think it would be helpful to relate the numbers of detected species to the number of known species of that taxon. This would put statements such as ‘most richly infected genus’ in perspective. Although in the rebuttal you claim these numbers are in the paper, I can’t find them anywhere. If you disagree with my view that this would be helpful, please explain. This is another point I have already mentioned in my first review.

L488–491 You say there’s only 3 studies, but then cite 8?


List of beetle genera with reports of Wolbachia not mentioned in the paper:
Acanthoscelides, Acoma, Acromis, Adoretus, Aeolesthes, Aeoloderma, Agelasa, Agra, Allonyx, Alphitobius, Anadastus, Anaspis, Anatis, Ancylopus, Anisosticta, Anomala, Anthonomous, Apalochrus, Aphidecta, Aslamidium, Asphaera, Aspidomorpha, Astycus, Aulacophora, Basilepta, Blosyrus, Bolbocerastes, Bolitotherus, Brachysomus, Brentus, Bruchidius, Bruchus, Brumoides, Byctiscus, Calomyterus, Calypteron, Calyptocephela, Canthidium, Canthon, Carpophilus, Centricnemus, Cephaloleia, Cetonia, Chauligognathus, Chersinellina, Chlaenius, Chrysolina, Cicindela, Cis, Clytus, Coelophora, Colaspis, Coleomegilla, Colpodes, Coptocycla, Coptodear, Cossonus, Cryptoglossa, Cryptolaemus, Cryptolestes, Cyclocephala, Cycloneda, Cycnotrachelus, Cypherotylus, Deloyala, Delphastus, Dicladispa, Dicranoncus, Diplotaxis, Disonycha, Drypta, Echinocnemus, Enoploplactus, Epilachna, Eubrianax, Eucryptorrhynchus, Eugnathus, Exochomus, Formicomus, Foucartia, Galerita, Gallerucida, Geotrupes, Germarostes, Glaresis, Gnathocerus, Grammoptera, Guignotus, Haptoncus, Harpalus, Hippodemia, Hotaria, Hybosorus, Hydrochus, Hydroglyphus, Hydronomidius, Hydrophilus, Imatidium, Larinus, Lema, Leptinotarsa, Leptura, Ligyrus, Liparus, Listrus, Lixus, Lobiopa, Lucanus, Lycostomus, Lygaria, Medythia, Megalocaria, Megalodacne, Micraspis, Monolepta, Mordellistena, Morion, Myrrha, Myzia, Neoglanis_or_Donus, Nicrophorus, Notoxus, Omorgus, Oncocerna, Onthophagus, Ophionea, Oryzophilus, Oulema, Oxythyrea, Pachyschelus, Palorus, Paracycnotrachelus, Parathyce, Passalus, Pentagonica, Peridinetus, Phelypera, Phrixopogon, Phyllobius, Phyllophaga, Phyllotreta, Platyphora, Pleocoma, Pogonocherus, Polychalma, Propylaea, Prosopodonta, Pselaphacus, Pseudimatidium, Pseudopthalmus, Psyllobora, Rhagonycha, Rhaphuma, Rhinostomus, Rhynchophorus, Rhyzobius, Rhyzopertha, Scarabaeus, Scarites, Sciaphilus, Scymnus, Semiotus, Serangium, Serica, Smaragdina, Spaethiella, Stegotes, Stenopterus, Stenotarsus, Sthenias, Subcoccinella, Submera, Sueus, Tapinaspis, Temnochila, Tenebrio, Tenebroides, Tetraopes, Trachyaphthona, Trogoderma, Typocerus, Tytthaspis, Xestolabus, Zigia

Reviewer 3 ·

Basic reporting

The authors have significantly improved the paper and adhered to most points raised in the review. The paper can be accepted almost as is, just some minor mistakes or inconsistencies caught my attention upon rereading it.


minor comments:
l. 116 - 'aspects of research' would be better than 'groups of research'
l.326 - 'both Chrysomelidae' - what does that refer to?
l. 339 - rearrange to 'Diabrotica virgifera virgifera on maize'
l. 403 - delete 'for'
l. 464 - the singular of bacteria is bacterium; otherwise put your verbs in plural
l. 469 - et al. misplaced
l. 501 - only three studies have found ..... the number of references indicated is far higher

Experimental design

Has been improved by redoing the literature search on WoS.

Validity of the findings

no further comments

---

## Round 0.3 · accepted · Accept

Dear Kajtoch:

Thanks for submitting your revision...I am happy to report that your work is now suitable for publication in PeerJ. Congratulations! Regarding the concern by Reviewer 2 about missed host associations, I would like you to make a brief statement about what you encountered with NCBI and the difficulty confirming the many putative host associations reported there - you can do this while in production. Other than that, I believe your manuscript is ready to go. Great job! I look forward to seeing your work in print! Thanks again for publishing with PeerJ!

-joe